# Sustainable Pest Management in Date Palm Ecosystems: Unveiling the Ecological Dynamics of Red Palm Weevil (Coleoptera: Curculionidae) Infestations

**DOI:** 10.3390/insects14110859

**Published:** 2023-11-06

**Authors:** Hassan Naveed, Vivian Andoh, Waqar Islam, Liang Chen, Keping Chen

**Affiliations:** 1School of Food and Biological Engineering, Jiangsu University, Zhenjiang 212013, China; hassan.naveed88@outlook.com (H.N.); vandoh@ujs.edu.cn (V.A.); 2School of Life Sciences, Jiangsu University, Zhenjiang 212013, China; 3State Key Laboratory of Desert and Oasis Ecology, Xinjiang Institute of Ecology and Geography, Chinese Academy of Sciences, Urumqi 830011, China; ddoapsial@yahoo.com

**Keywords:** date palm, insect activity, pest detection, biology, integrated pest management, environment

## Abstract

**Simple Summary:**

This summary explores strategies to protect date palm trees from harmful insects like the red palm weevil. The emphasis is on employing intelligent and eco-friendly methods that do not harm the environment. The goal is to understand the impact of these insects on trees and identify effective measures to prevent their infestation. Ultimately, the objective is to ensure the long-term health and vitality of date palm trees.

**Abstract:**

The red palm weevil (RPW) poses a significant threat to date palm ecosystems, highlighting the necessity of sustainable pest management strategies that carefully consider the delicate ecological balance within these environments. This comprehensive review delves into innovative approaches to sustainable pest management, specifically focusing on date palm, and seeks to unravel the intricate ecological dynamics underlying RPW infestations. We thoroughly analyze biocontrol methods, eco-friendly chemical interventions, and integrated pest management (IPM) strategies, aiming to minimize the ecological impact while effectively addressing RPW infestations. By emphasizing the interplay of both living organisms (biotic) and environmental factors (abiotic) in shaping RPW dynamics, we advocate for a holistic and sustainable management approach that ensures the long-term resilience of date palm ecosystems. This review aims to contribute to an ecologically sound framework for pest management, promoting the sustainability and vitality of date palm ecosystems amidst the challenges posed by the RPW.

## 1. Introduction

The RPW, scientifically known as *Rhynchophorus ferrugineus* (Olivier, 1790), is a persistent pest that originated in Southeast Asia and Melanesia. Through accidental introduction, it has become a global threat, affecting regions such as the USA, the Caribbean, the Mediterranean basin, and the Middle East. The RPW has been observed in numerous regions, spanning 28 countries in Asia, 6 in Africa, 1 in North America, 2 in Central America and the Caribbean, 14 in Europe, and 5 in Oceania [1,2]. The RPW exhibits the widest global distribution among the *Rhynchophorus* weevil genus and is notorious for causing extensive damage to cultivated date palm farms. Its existence has resulted in notable challenges concerning the cultivation of palm and coconut trees and their use in ornamental settings [3,4]. This invasive species has targeted about 40 different species of palm trees globally [5], resulting in its classification on the A2 list along with a few other destructive pest species, including *Bemisia tabaci*, *Drosophila suzukii*, *Helicoverpa armigera*, *Spodoptera frugiperda*, *S. litura*, and *Trogoderma granarium*, by the European and Mediterranean Plant Protection Organization (EPPO, 2002) [6]. While the majority of affected palms belong to the Arecaceae family, there have been reports of infestations in one palm species each from the Agavaceae and Poaceae families as well [5,7,8]. The impact of the RPW on date palm (*Phoenix dactylifera*) cultivation is substantial. In the Arabian Gulf region alone, it caused a staggering loss of US$25.92 million in 2009, accounting for approximately 30% of the world’s annual date palm production [9]. Unfortunately, infestation symptoms often go unnoticed until extensive damage has occurred, posing challenges for effective intervention once detected. As a result, early detection of RPW larvae is crucial to prevent their infiltration into the vascular system of date palms and the subsequent initiation of infestations. Taking swift action at this stage is essential to mitigating the damage caused by the pest.

The RPW presents a significant invasive threat and may exhibit cryptic behavior, as observed in studies by Giblin-Davis et al. [10], Mahmud et al. [11], and Pontikakos et al. [12], allowing it to infiltrate uninfested areas without detection. Despite the implementation of the recommended management measures, controlling outbreaks of the RPW has proven to be ineffective [13,14]. The RPW has a predator/prey relationship with palm trees throughout its life cycle. The larvae of the RPW feed on the trunks of palm trees, causing significant damage and posing a threat to both wild and cultivated palms in various countries. When date palms are infested by the RPW, various symptoms may appear, depending on the stage of the attack. These symptoms include the secretion of a brown fluid with a fermented odor, resulting from a combination of palm tissue and larval feeding secretions. Injured trees emit highly volatile compounds known as kairomones, which attract male weevils [15]. Further indications of infestation encompass the creation of tunnels within the palm tissue, the appearance of adult and pupal stages at the frond base, desiccated offshoots, pupae clustering around the palm base, wilting outer leaves, and, in severe instances, the complete or partial loss of the upper trunk due to extensive tissue damage [16]. The widespread occurrence of invasive RPW populations has raised concerns about the economic and ecological consequences, prompting global efforts from governments, companies, and researchers to develop effective control strategies and minimize further damage. Addressing the challenges associated with agricultural pest management, such as climate change, increasing insecticide resistance, and rising food production demands, necessitates a thorough understanding of RPW biology. Furthermore, it explores the application of advanced technologies, including transcriptomics, genomics, and metagenomics, in studying the RPW [17]. By capitalizing on the knowledge gained using these powerful tools, researchers can enhance their understanding of the RPW’s biology and behavior, thereby enabling the development of more effective management strategies. The ultimate goal of this review is to contribute to collective efforts in combating the RPW threat and ensuring the sustainable management of palm trees.

There is a similar species affecting oil palm cultivars, the black weevil, *R. palmarum*, that causes a very serious problem to oil palm cultivars and coconuts in Latin America, South America, and the Caribbean [18]. In Colombia, this insect is widely distributed and constitutes an important phytosanitary problem due to the damage caused to oil palms. The damage can be direct or indirect and, in both cases, causes the death of the palms [19]. *R. palmarum* is an insect involved with the diseases of the oil palm, Red Ring (RR) and Bud Rot (BR). It is a vector of the nematode *Bursaphelenchus cocophilus* (Cobb) Baujard. They acquire the nematode during its larval stage, while it develops in the palm, or in the adult stage, by feeding on contaminated tissues. The nematode can be found in larvae, pupae, and adults, both internally and externally. In larvae, it is found in the intestine, hemocele, and tracheae; in adults, it is in the intestine, the body cavity, and the ovipositor [20].

In the adult stage, *R. palmarum* is attracted to the fermentation of exposed tissues, such as wounds or cuts on the leaves or palms diseased with BR. If the adult is contaminated, it can inoculate and infect the palm by feeding on or ovipositing in these tissues [21].

Several techniques have been developed to detect infestations of the RPW in palm trees. These include visual inspection [22], utilization of acoustic sensors [22,23], specially trained dogs with olfactory capabilities [24], trapping [25], canopy inspection [26], endoscope inspection [27], drones and aerial surveys [28], biochemical analysis [14], soil analysis [29], and X-ray CT (computed tomography) scans [30,31]. However, some methods have demonstrated limitations in terms of their accuracy. Detecting the presence of the RPW within tree trunks presents challenges that can impact the reliability of the results. Early detection of infestation presents a challenge as it is typically identified after the palm tree has already suffered extensive damage. The weevil’s destructive behavior is intensified by certain conventional agricultural practices, such as leaf removal during harvesting or pruning of branches, inadvertently causing harm to the trees. Currently, the primary strategy for RPW control predominantly depends on the utilization of synthetic chemical insecticides. However, mounting concerns about the environmental pollution associated with these treatments have emerged.

Recent studies have provided evidence supporting the correlation between the ability of plants to resist or tolerate insect pests and diseases and the optimal physical, chemical, and primarily biological characteristics of the soil and plants themselves. The resistance of plants is directly linked to their physiological attributes, and any factors influencing plant physiology can potentially alter their resistance to insect pests [32]. Numerous insecticides have undergone testing in laboratory and field experiments worldwide to combat RPW infestations. These include azadirachtin, spirotetramat, methidathion, zinc sulphate, essential oils derived from *Eupatorium adenophorum* and *Artemisia nilagirica*, as well as oxamyl, which target all stages of the RPW.

This review aims to comprehensively understand the biology of the RPW, including its life cycle, feeding habits, behavior, and reproductive patterns. It also intends to assess the impact of RPW infestations on date palm ecosystems, taking into account ecological, economic, and social aspects like tree damage, fruit yield reduction, and economic losses. The primary objective is to categorize and evaluate various sustainable pest management strategies for RPW control. These strategies encompass biological, cultural, physical, and chemical methods, all emphasizing eco-friendly and cost-effective approaches. Additionally, the review investigates how environmental factors such as climate, soil conditions, and habitat structure influence RPW population dynamics. This provides insights to tailor location-specific management strategies. Furthermore, the review critically assesses the effectiveness and integration of different components of IPM in handling RPW infestations. It sheds light on success stories, challenges, and future prospects. This advocacy promotes a balanced ecosystem and reduces reliance on chemical control methods. It consolidates existing knowledge, guides sustainable practices, fosters innovation, and addresses a critical agricultural challenge that directly impacts global food security and environmental sustainability.

## 2. Life Cycle and Behavior of the RPW

The genus *Rhynchophorus* (Coleoptera: Curculionidae) consists of oligophagous insect pests that reproduce on a wide range of palm species (Arecales: Arecaceae), undergoing complete metamorphosis [33]. The majority of the RPW’s life cycle occurs within the palm tree itself. Females excavate an egg chamber, typically at the base of a palm frond or in a damaged plant area. They possess the capacity to lay 200–300 eggs over their lifespan [34]. Additionally, the RPW can continue to grow during the winter as long as undamaged tissue remains at the outer part of the trunks. The insect has the capacity to go through three to four generations within a single palm tree in a year.

The eggs are oblong, creamy white, shiny [35], and individually deposited into palm sheaths and stems [36]. The eggs typically hatch within 2–5 days, giving rise to legless neonate larvae [37]. The larvae have a yellowish-white coloration with a brown head and possess a hard cuticular head capsule that aids in food consumption [38]. These larvae possess mandibles that are heavily chitinized and maxillae that are less sclerotized. The mandibles resemble pinchers with cutting edges and teeth, enabling them to grip, crush, and cut palm tissues. Conversely, the maxillae possess mechanoreceptors and chemoreceptors that detect foods such as soft palm tissues and palm liquid before consumption [39]. RPW larvae typically burrow into palm trees, starting from their egg-laying sites. The apical meristem of the palms is critical for the development of leaves and fruits [40]. Therefore, significant damage to the leaves of RPW-infested palms can have adverse effects on photosynthesis and overall palm growth. Although the larval stage lasts the longest, with 10–13 instars inside the palm tree tissues, typically ranging from 35 to 100 days, RPW larvae do not continuously feed [41].

During the pupation stage, RPW larvae discontinue their feeding activities and commence the construction of a cocoon. The cocoon has an oval shape and is constructed using palm fibers. The head of the RPW pupae bends downward (ventrally) and has a long rostrum with noticeable antennae and eyes. Initially, RPW pupae have a creamy coloration. The pupation process typically takes around 13–17 days, after which adult weevils emerge. The emergence of RPW adults is influenced by the ambient temperature [42]. The entire life cycle of the RPW, from egg to adult, lasts about 4 months. 

Once they emerge from their cocoons, adult RPWs demonstrate the ability to fly considerable distances [43]. Male RPWs release pheromones to attract females for mating [44]. RPWs release volatiles while infesting palm trees, aiding in their clustering [45]. Morphological identification of RPWs relies mainly on the features of adult individuals [30]. Both male and female RPWs have a reddish-brown cylindrical body with an extended rostrum. The dorsal side displays a reddish-brown color, while the ventral side appears dark brown. RPWs possess elongated mouthparts, forming a slender snout with a pair of mandibles at the end and a pair of antennae near the base. One of the simplest methods to distinguish between male and female RPWs is by examining their snout characteristics. In females, the rostrum is bare, slender, curved, and a little longer than in males [34]. The adult RPW typically exhibits a reddish-brown body color but may also appear dark with a red streak. Notably, RPW adults display significant phenotypic diversity, with body colors ranging from entirely orange-red to all black, including various intermediate patterns with different numbers and sizes of black marks. To accurately identify the species, recent genetic studies have employed the mitochondrial cytochrome oxidase subunit I (COI) gene, which proves to be the most reliable method [46].

## 3. Integrated Pest Management: A Holistic Approach to Sustainable Control of the Red Palm Weevil

Adapting IPM to various geographical regions and palm tree species requires tailoring approaches to suit specific ecosystems. Accessibility to advanced technologies and the costs associated with their implementation pose challenges, especially in developing regions. Continued research and development in biotechnology and precision technologies will enhance the precision and efficiency of RPW management. Collaborative efforts and knowledge sharing among researchers, governments, and stakeholders can lead to the widespread adoption of IPM globally. Tailoring IPM strategies to unique ecological and geographical contexts ensures optimal pest control outcomes. However, technological inequalities, financing restrictions, and poor infrastructure, particularly in areas with few resources, prevent the widespread use of IPM. Bridging this gap requires targeted investments in research, innovation, and capacity building, emphasizing cost-effective and accessible technologies for RPW control. Advancements in biotechnology, including the development of RPW-resistant palm varieties and precise pest detection methods, hold promise for revolutionizing IPM strategies (Figure 1). Integrating these innovations globally can significantly bolster the efficiency and sustainability of RPW management. This collective approach promotes the sharing of best practices, the dissemination of research findings, and the development of region-specific IPM guidelines, accelerating the widespread adoption of effective RPW management strategies on a global scale. 

IPM offers a sustainable approach for managing the RPW and safeguarding palm ecosystems. By integrating monitoring and surveillance, biological control, cultural practices, and targeted chemical control with precision technologies, IPM maximizes effectiveness while minimizing environmental impact. With ongoing advancements in precision tools and technologies, IPM holds immense promise for effectively controlling RPW infestations and ensuring the longevity and health of palm trees. Collaboration and investment in research and implementation are key to realizing the potential of IPM for the sustainable control of the RPW and the preservation of palm ecosystems.

### 3.1. Cultural Control Methods: Sanitation, Pruning, and Removal of Infested Palms

Managing the RPW effectively requires prioritizing field and crop sanitation to eliminate potential breeding sites. In cases where palm trees are severely infested and cannot be saved, the recommended approach is to destroy them using shredding. Burning is not a suitable option because green palms are resistant to burning, allowing weevil stages deep inside the tree to survive. As part of phytosanitation practices, it is crucial to treat freshly cut or injured palm surfaces with insecticide after trimming the fronds or removing the offshoots. This approach is crucial in removing palm volatiles that might attract female weevils seeking appropriate locations to lay their eggs [47].

For effective cultural management of RPW infestations, consider the following recommended practices: regularly remove and dispose of fallen or damaged palm fronds to prevent them from becoming breeding grounds and food sources for weevils; trim palm trees by removing dead or dying fronds to minimize potential breeding sites for weevils; install pheromone traps in and around palm trees to attract and capture adult weevils; wrap the trunks of palm trees with materials such as hessian or adhesive bands to trap climbing weevils; administer systemic insecticides directly into the trunks of palm trees to target and control weevil populations; when choosing palm tree species or cultivars, opt for those that are less susceptible to RPW infestation; ensure the overall health and vitality of palm trees via the proper watering, fertilization, and general tree care practices. By implementing these strategies, it is possible to effectively manage RPW infestations and maintain the health of palm trees. Successfully cultivating and managing date palm trees demands growers’ careful attention, as their well-being and productivity directly link to the practices employed. Various field operations play a pivotal role in ensuring the health of date palm trees and mitigating pest infestations. These operations encompass important tasks such as offshoot selection, appropriate spacing, optimal fertilization, efficient irrigation techniques, fruit thinning, leaf pruning, and effective harvesting methods [48]. In evaluating the efficacy of field operations, Al-Shawaf et al. [49] validated an IPM program for the RPW in the Al-Ahsa oasis of Saudi Arabia (SA). The success of the program was measured based on the data gathered from palm removal, with the findings indicating that eradicating more than 20% of infested palms is undesirable, signifying the need for adaptive control strategies when the pest population persists. An earlier study provided a detailed protocol for the safe removal and disposal of severely infested date palms [5]. This protocol involves identifying and marking severely infested or damaged palms using colored tape, spray paint, or designated straps. A severe infestation is identified when over 30% of the trunk tissue is damaged at the infestation site. Initiating the removal process promptly is crucial to prevent the spread of adult weevils from infested palms to nearby healthy ones, thereby simplifying the task. As a preventive measure, the application of the recommended pesticides via soaking, drenching, or showering the palm crown, trunk, and bole regions is undertaken. In specific regions, shredding machines are utilized to eradicate severely infested palms at specified sites. The infested palms undergo cutting into logs, after which various palm components such as fronds and trunks are transported to the shredding location [50]. By adhering to these guidelines, growers can effectively manage pest infestations, preserve the health of date palm trees, and safeguard the overall production of date palms (Figure 2). When cultivating perennial crops like palms, farmers often choose commercially popular and well-established cultivars that are commonly grown in their regions. However, these favored cultivars are frequently highly susceptible to infestation by the RPW. Although some initial research has investigated the tolerance or susceptibility of different palm cultivars to the RPW [51,52,53], there is still a significant knowledge gap regarding the exploration and utilization of host plant resistance against this pest.

To bridge this knowledge gap, it is crucial to devise screening techniques capable of identifying RPW-resistant cultivars and selecting suitable parental materials for breeding programs. Additionally, leveraging advanced molecular techniques such as marker-assisted breeding and RNA interference (RNAi) can expedite the development of RPW-resistant cultivars [54]. Regarding agro-techniques for RPW management, several factors play a pivotal role. These factors encompass optimizing palm density, implementing appropriate irrigation methods, and promptly protecting the tissue after frond and offshoot removal. These practices have been recognized as critical elements in effectively mitigating RPW infestations [48,53,55].

### 3.2. Biocontrol Marvels: Exploring Natural Predators (NPs) for Red Palm Weevil Regulation

The process of managing RPW infestations involves several key steps including introducing predatory insects (e.g., *Anisolabis maritima*, *Chelisoches morio*, *Platymeris laevicollis*, *Xylocorus galactinus*, and *Scolia erratica*) and nematodes (e.g., *Heterorhabditis bacteriophora*, *Praecocilenchus ferruginophorus*, and *Steinernema glaseri*), mass rearing and increasing their populations, conducting biological monitoring, conserving natural enemies, and continuously researching and developing new potential predators and nematodes for effective control. It is crucial to consider the local conditions, the effectiveness of control measures, and their compatibility with other methods when selecting and implementing biological control strategies. This approach aims to achieve successful and sustainable management of RPW infestations.

Various countries have extensively researched biological control methods for managing RPW infestations [37,56]. Noteworthy studies include the research conducted by Hussain et al. [57] in SA, which investigated the effectiveness of different sesquiterpenes against the RPW. One of the sesquiterpenes, Picrotoxin, exhibited significant toxicity with an LD50 value of 317 ppm. Similarly, AlJabr et al. [58] reported the larvicidal and growth-inhibiting activities of phenylpropanoids against the RPW in SA. Hajjar et al. [59] focused on the use of commercial formulations of the *Beauveria bassiana* (Balsamo-Crivelli) [60] fungus for effective biological control of adult RPWs in SA. Hussain et al. [61] evaluated the larvicidal activity of various concentrated extracts of *Piper nigrum* seeds against the RPW. Al-Deeb et al. [62] documented the presence of three phoretic mites belonging to the genera *Uropoda*, *Uroobovella*, and *Curculanoetus* on the RPW in the UAE and suspected these may affect the weevils’ health. Francesca et al. [63] investigated the ovicidal and larvicidal activities of entomopathogenic bacteria against the RPW [63]. Abdullah [64] conducted research on the toxicity and pathological effects of two natural biopesticides, *Boxus chinensis* oil and precocene II, on RPW larvae. Further studies have demonstrated the potential of *S. carpocapsae* in a chitosan formulation in controlling RPW infestations [65]. Abdel-Samad et al. [66] found that a commercial oil formulation of *B. bassiana* exhibited significant toxicity against the RPW. *E. adenophorum* and *A. nilagirica* essential oils were identified as having notable antifeedant activity against the RPW [67]. Moreover, *P. nigrum* demonstrated excellent insecticidal activity against the RPW [61]. Pu et al. [68] highlighted the efficacy of Bacillus thuringiensis, an entomopathogenic bacterium strain, as a biological control agent against the RPW. The maritime earwig, *A. maritima* (Dermaptera: Carciniphoridae), has been documented as a predator of RPW eggs and newly hatched larvae [54]. Several isolates of entomopathogenic nematodes (EPNs) have been found to infect adult RPWs but a few causing the highest mortality of RPWs include *H. bacteriophora*, *H. indicus*, *S. abbasi*, *S. carpocapsae*, *S. feltiae*, and *S. riobrave* [8].

In their study, Mazza et al. [56] provided a comprehensive list of more than 50 biological control agents that have proven effective against weevils from the *Rhynchophorus* group. These included viruses (1 species), bacteria (8 species), fungi (9 species), yeasts, nematodes (7 species), mites (11 species), insects (12 species), and vertebrates (6 species). Among these, fungal biocontrol agents were found to be the most effective choice. They meet the criteria for biocontrol in various aspects and are well suited to different contexts [56]. The parasitoid mite *Rhynchopolipus rhynchophori* has shown the ability to reduce the RPW population density by feeding on their body fluids in laboratory conditions [64]. However, further research is needed to determine whether these mites have a pathological effect on the weevils. Yasin et al. [69] conducted an extensive review on the use of microbial agents for controlling RPWs and highlighted the promising potential of the *B. bassiana* and *Metarhizium anisopliae* strains, specifically isolated from naturally infected RPWs, as effective biological control agents for this pest. Many studies reveal that EPNs and entomopathogenic fungi (EPF) work well in labs and semi-field tests, making them good options to control RPWs [8,55,56,57]. The mass production and storage of EPNs as biopesticides can be achieved using two methods: in vivo and in vitro [69]. EPF like *M. anisopliae* and *B. bassiana* are commonly utilized in IPM strategies to control RPWs in field settings [70,71,72]. The pathogenicity of indigenous isolates, particularly the *M. anisopliae* strain MET-GRA4, has been investigated in vitro, revealing high efficacy with a 100% mortality rate observed 21 days after the infection of adult RPWs [73]. Controlling infestations of the RPW in palm trunks can be effectively achieved via the application of biological control agents. Concerning viral control, the cytoplasmic polyhedrosis virus (CPV) has been identified as the sole virus found in RPWs. Initially found in Kerala, India, this potent virus infected RPWs at different growth stages. Infections during the later larval stage led to the emergence of deformed adults and a notable decline in insect populations [56,74].

The effectiveness of *B. thuringiensis*, an entomopathogenic bacterium known for producing insecticidal crystal proteins, has been studied against both larvae and adults of the RPW. In lab tests, *B. thuringiensis* and *B. cereus* showed high efficacy in controlling RPW larvae and adults [75]. Another investigation using the nematode *S. carpocapsae* in combination with chitosan (Biorend R^®^) reported high larval mortality (>50%) at a concentration as low as 0.5 mg/mL, with mortality reaching 85% at 2.0 mg/mL [76]. Yasin et al. [77] observed larval mortality ranging from 46.86% to 58.36% and adult mortality ranging from 26.79% to 39.04% after 21 days of exposure to *B. thuringiensis*. Furthermore, the use of microwave-heating treatment has emerged as an alternative method to address RPW infestations without causing substantial harm to the host plant. By inducing hyperthermia in RPW adults and larvae, microwave radiation offers a safe and environmentally friendly method that only results in slight dehydration of the palm trees [78,79].

Plant extracts from various species, like French marigold (*Tagetes patula*) [80], Ceylon (*Cinnamomum zeylanicum*) [81], citronella grass (*Cymbopogon nardus*) [82,83], clove (*Syzygium aromaticum*), and cardamom (*Elettaria cardamomum*), have been effective in controlling RPWs due to their insecticidal properties [16]. This production capacity has led to the commercialization of at least 13 species of Steinernematids and Heterorhabditids for insect control purposes [84]. Various storage and transportation methods for EPNs have been employed, including aqueous suspension, synthetic sponges, gels, clay, powder, and infected cadavers [85]. These methods have been widely adopted and commercialized in several countries. For instance, Sanoplant in Switzerland, Helix in Canada, ORTHO Biosafe USA in the USA, and BASF in Germany have commercialized *S. carpocapsae* [85,86]. Numerous studies have substantiated the effectiveness of EPNs as biological control agents in managing RPW infestations. The symbiotic relationship between EPNs and their associated bacteria has proven to be significant, showing promising results in addressing the issue of RPW infestation.

### 3.3. Sustainable Chemical Approaches to Combat Red Palm Weevils

Understanding the life cycle of the RPW, from egg to adult, provides crucial insights into designing targeted chemical interventions. The life stages present unique vulnerabilities for targeted pest management. The economic and ecological impact of RPW infestations underscores the urgency of developing sustainable and efficient control strategies. Sustainable chemical approaches hold promise in combating the RPW while ensuring the preservation of the ecosystem. Through careful evaluation of biopesticides and chitin synthesis inhibitors, an eco-friendly arsenal can be developed to effectively manage RPW populations. Continuous research, innovation, and collaboration are essential to enhance the sustainability and efficacy of these chemical approaches, ultimately contributing to the conservation of palm trees and the ecosystems they support.

Chemical control is one of the methods commonly used to manage RPW populations (Figure 2). However, there are several challenges and considerations associated with chemical control strategies for this pest. Here are some key points to consider: selection of effective insecticides, timing of application, penetration of the insecticide, resistance management, environmental considerations, regulatory compliance, monitoring and evaluation, and IPM. In a recent study by Alhewairini [87], oxamyl was tested against the RPW, and the results were significant. After 1, 24, and 48 h, the adult mortality was 62, 82, and 100 percent, while the larval mortality was 72, 77, and 100 percent. However, it is crucial to consider that insecticides can be costly, excessive usage can have detrimental effects on the environment, and it may contribute to the development of chemical resistance in weevil populations [88]. In areas where RPW infestation is prevalent, preventive spray or shower applications of insecticides are commonly employed to protect date plantations and control the spread of the weevils. To prevent female weevils from laying eggs, it is recommended to apply insecticides directly to fresh injuries. In the early stages of infestation, curative insecticide treatments via stem injection techniques are effective in treating affected palm trees. Recently, modern insecticides such as neonicotinoids (imidacloprid) and phenylpyrazoles (fipronil) have been employed to prevent and treat RPW infestations in date palms [89,90]. In some countries, stem injection of infested date palms is carried out using pressure injectors, but it is important to exercise caution and ensure that the pressure does not exceed 1 bar to prevent permanent tissue damage when introducing the insecticide at higher pressures. It is crucial to conduct such treatments under the supervision of trained personnel [54].

Wakil et al. [91] tested how the RPW responded to common insecticides like profenophos, imidacloprid, chlorpyrifos, cypermethrin, deltamethrin, spinosad, lambda-cyhalothrin, and phosphine by using the diet incorporation method. The results revealed that the RPW displayed higher resistance to cypermethrin, deltamethrin, and phosphine, underscoring the prolonged impact of insecticide use. While synthetic pesticides have traditionally been relied upon for RPW control [92], their use is unsustainable and poses risks to the environment, biodiversity, and human health [93,94]. Additionally, Al-Ayedh et al. [95] have argued that synthetic pesticides are insufficient for effective RPW control. Consequently, there is growing demand for the adoption of IPM strategies to tackle the RPW [96,97]. IPM involves the integration of cultural techniques, ecosystem health techniques, and biological and chemical approaches to reducing pesticide dependency while maintaining crop yield, quality, and profitability [98,99,100].

Pheromone traps are essential to the RPW’s IPM. To effectively manage the RPW in date plantations, the use of pheromone traps is commonly employed (Figure 2). These traps can be situated at ground level, partially inserted into the soil, or hung from palm trunks. However, it is advisable to avoid hanging traps on the trunk for as long as possible, as extended periods may have adverse effects on the health and integrity of the palm trees. To improve the efficiency of aggregation pheromone traps in combating the RPW, it is crucial to employ black traps (featuring openings and funnels and without covers) positioned at ground level [101,102]. Bucket traps have been used for beetles and weevils. It has been found that larger traps with 9–10-L plastic buckets demonstrated better results than smaller ones [103]. Various factors, including trap design, placement, and the specific pheromone used, can shape how pheromone traps influence other arthropod groups. While pheromone traps are typically designed to target specific pests and minimize their impact on non-target species, unintended consequences can still occur. Research suggests that pheromone traps may have the potential to attract non-target insects, inadvertently leading to their capture. The organisms of particular concern include predators and parasitoids of target species or other pests; pollinators, with a specific emphasis on honey bees; arthropods, with significance in conservation; and aquatic organisms [104]. It is important to note that many insect predators, such as lady beetles, hoverflies, and minute pirate bugs, along with parasitic wasps and honey bees, have displayed preferences for specific trap color characteristics. In contrast, green lacewings and spiders did not exhibit such preferences [105]. Placing the traps in shaded areas is recommended to maximize their effectiveness and prolong the lure’s lifespan. The pest density in the field typically determines the quantity of traps needed for mass trapping programs, usually falling within the range of 1 to 10 traps per hectare [106]. Mass trapping of adult RPWs using pheromone traps baited with food is a crucial component of their management [37]. The primary aggregation pheromone used for RPWs is ferrugineol [107,108], often supplemented with 4-methyl-5-nonanone, a ketone often found in certain fruits and plants, in mass trapping efforts conducted in affected countries [5,109]. These pheromone traps have a higher attraction for female weevils compared to adult males, resulting in an average capture ratio of around two females for every male weevil [107,110,111]. This finding holds great importance in RPW management since capturing female weevils before they commence egg laying helps prevent the establishment of new infestations.

To enhance the efficiency of pheromone traps, Al Ansi et al. [112] evaluated how trap location, temperature, degree of palm fruit fermentation, and pheromone lure source affect RPW capture rates. Insects such as the RPW depend on semiochemicals, which are chemical signals that aid in communication and interactions. Semiochemicals can be divided into two groups based on the sender and receiver of the message, with pheromones being the chemical signals conveying information within the same species. Pheromone traps are integral to IPM strategies for the RPW. Previous studies on the use of pheromone traps in the IPM of the RPW have focused on lure sources and pheromone release rates [113,114], trap design [109], trap color [102,115], the number of traps per hectare [106], and kairomones [108]. The number of RPWs caught in pheromone traps is influenced by the trap placement (at the palm orchard’s edge or middle), the temperature of the surroundings, the origin of pheromones, and the level of date fruit fermentation (acting as kairomones) [112].

The choice of pheromone lure source had a noticeable impact on the capture rates of traps. Ferrolure traps exhibited higher capture rates for adult RPWs compared to traps using Rhylure. However, when evaluating the effectiveness of four commercial lures, all lures were equally successful in attracting adult RPWs in both field and laboratory tests [116]. Similarly, no significant differences in the number of captured RPWs per week were observed when utilizing three commercial pheromone lures (Ferrolure+™, RHYFER™, and RHYNCAP™) [117]. Additionally, Y-tube olfactometer tests revealed a relatively higher attraction of females to the traps and volatiles. This could be attributed to the presence of a greater number of olfactory sensilla on the antennae of female RPWs.

## 4. Role of Ecosystem Health in Red Palm Weevil Population Dynamics

Given the significant economic and ecological damage caused by the RPW on a global scale, it is crucial to grasp how ecosystem health influences the population dynamics of this destructive pest. This understanding is essential for devising effective management strategies. Understanding how the environment, biology, and human actions affect RPW populations is very important. Diverse elements of ecosystem health, such as biodiversity, habitat structure, climatic conditions, and human activities, shape RPW population dynamics and offer valuable insights for the pursuit of sustainable pest management.

Biodiversity plays a crucial role in regulating RPM populations as healthy ecosystems with diverse flora and fauna can provide natural predators and competitors that help keep its populations in check [118]. Changes in biodiversity, whether due to host plant variability, predator–prey dynamics, ecosystem stability, or invasive species, have profound impacts on RPW populations. By analyzing these dynamics, we can develop strategies that promote biodiversity and harness its potential to mitigate the harmful effects of RPW infestations on palm ecosystems [119]. Understanding these relationships is vital for predicting and mitigating pest outbreaks effectively.

Climate, including temperature, rainfall, and seasonal variations, plays a critical role in the distribution and abundance of RPWs. Climate change and extreme weather events may affect RPW population dynamics and spread, emphasizing the need for climate-adaptive pest management strategies [120]. Climate change has a detrimental impact on the distribution and abundance of RPWs [121]. A study conducted by Hussain et al. [122] revealed that climatic factors, particularly daily mean temperature and relative humidity, significantly influence the abundance of RPW adult populations. Dembilio et al. [123] led research that unveiled the pivotal role of temperature in influencing RPW’s reproductive parameters, such as oviposition and egg hatching. This impact is particularly pronounced in regions with mean monthly temperatures falling below winter averages, a common occurrence in most Mediterranean areas [123]. These findings highlight the potential for climate change to reduce RPW abundance and disrupt its distribution, particularly in tropical and marine climates [124].

El-Lakwah et al. [125] have identified a positive correlation between RPW population abundance and average temperature, while relative humidity exerts a negative influence. These climatic factors also affect the emergence and flight patterns of RPW adults, resulting in peak populations during specific times of the year [125]. Furthermore, research by Cinnirella et al. [126] has proposed a spatial spread model to elucidate the dynamics of RPW spread. This model suggests that the weevil initially occupies as much space as possible during the early stages of infestation and subsequently increases in density as it colonizes new areas [126]. Higher temperatures create an environment suitable for the introduction and establishment of pests. This can aid in the establishment of invasive plant pests that might not have flourished otherwise. Moreover, the globalization of markets and modern transportation systems in recent times has created advantageous conditions for pests to move, invade, and establish themselves worldwide [118]. Human activities, such as international trade, transportation, and inadequate waste management, contribute to the spread and establishment of RPWs in new regions. IPM involves utilizing a combination of pest control measures while minimizing the impact on the environment. Understanding the ecosystem’s role in RPW population dynamics is critical for developing and implementing effective IPM strategies. An ecosystem-based approach can enhance the sustainability and efficiency of RPW management [17].

## 5. Challenges of RPW Management and Future Prospects

Managing the RPW poses several challenges, and there are ongoing efforts to improve control methods and develop future prospects for its management. Some of the challenges faced and potential future prospects include limited control options; while cultural control methods, chemical insecticides, and biological agents are available for managing the RPW, none of these methods provide a foolproof solution. The weevils have developed resistance to some insecticides, and biological control agents may not be widely available or effective in all situations. Developing new control options and integrated approaches is crucial. Early detection and monitoring of RPW infestations is challenging since the weevils remain hidden within the palm trees. Monitoring techniques, such as pheromone traps and visual inspections, have limitations in terms of accuracy and efficiency. Advancements in detection technologies, such as remote sensing, drones, and trained dogs, could improve the early detection and monitoring of weevil populations. The RPW is highly mobile and can spread rapidly over long distances. Human-mediated transportation of infested plant material is a significant factor in its spread. Strict quarantine measures and awareness campaigns are necessary to prevent the introduction of the weevil into new areas. International collaborations and regulations can help minimize the risk of spread. Educating the public about the weevil’s biology, signs of infestation, and appropriate management practices is crucial to ensure early detection and the implementation of control measures. Future prospects for the management of the RPW include genetic approaches. Genetic research and breeding programs could focus on developing palm tree varieties with natural resistance or tolerance to RPW infestation. Identifying the genetic markers associated with resistance traits can facilitate the breeding process. There is ongoing research to develop effective biopesticides derived from microbial pathogens, such as EPNs, fungi, and bacteria, specifically targeting the RPW [8,127]. These biopesticides offer environmentally friendly alternatives to chemical insecticides. Further advancements in pheromone-based trapping systems and attractants can enhance the efficiency of monitoring and mass trapping programs. Research is ongoing to develop more attractive synthetic pheromones and optimize trap designs.

Before the scientific consultation and high-level meeting on RPW management in Rome in March 2017, the Food and Agriculture Organization (FAO) worked on creating an informative document (FAO, 2017) (http://www.fao.org/3/a-ms664e.pdf) (accessed on 23 September 2023) [128] that presents an overview of the existing situation of the RPW in the Near East and North Africa (NENA) region. This document encompasses an outline of the current management practices, identification of challenges and weaknesses, and exploration of available research and technologies for potential improvement. However, it is evident that there are several gaps and challenges within the fundamental elements of the current IPM strategy for the RPW. The effectiveness of the current RPW IPM programs, which heavily rely on techniques such as pheromone trapping, has shown limited success. Challenges and gaps are evident in different aspects of the strategy, particularly in the early detection of pests, the development and implementation of phytosanitary measures, the effectiveness of biological control agents in field conditions, farmer participation in the programs, and the lack of socio-economic data, among other factors. These challenges collectively contribute to the significant difficulty in effectively controlling and eradicating the RPW. Nevertheless, noteworthy achievements have been made, such as the successful eradication of the pest in the Canary Islands and the ongoing progress toward eradication in Mauritania. The foundational document prepared by FAO is a valuable resource that provides insights into the present state of RPW management in the NENA region. It sheds light on the existing gaps and challenges that require attention and serves as a basis for discussions and the formulation of strategies to enhance RPW control and management in the future.

Efforts to eradicate the RPW have yielded success in various oases; however, the introduction of infested palms has undermined these achievements [129]. One of the primary challenges at the initial stage of the strategy is to provide farmers with a cost-effective and efficient device for early detection. Numerous laboratories worldwide have explored advanced techniques, including chemical signature detection, acoustic detection, infrared cameras, thermal imaging, satellite imaging/IoT, and more [22,130,131]. Nonetheless, visual inspection of palms remains a crucial method for identifying RPW-infested trees. In the scenario of using pheromone traps, a significant drawback is the requirement for regular upkeep of food-baited traps, entailing the changing of bait and water and maintaining records of weevil captures. Some progress has been made in addressing this challenge by using trap- and bait-free trapping (attract and kill) and dry trapping (Electrap™). However, there is still a need to improve data collection methods for weevil captures. Ideally, a dry trap that can automatically record and transmit weevil capture data would be a promising development for the future. Apart from attract and kill methods, other semiochemical-based control approaches to the RPW, such as the ‘push–pull’ technique involving the use of repellents and attractants [132] and the ‘attract and infect’ method involving the dissemination of biological control agents (EPFs) using pheromone traps [71], require further refinement and research.

Dependence on chemical insecticides for both prevention and treatment underscores the significance of exploring alternative approaches. Essential research into the effectiveness of natural insecticides against the RPW is crucial, advocating for their adoption and seamless integration into the control strategy. Well-managed plantations have demonstrated that regular calendar-based preventive insecticidal treatments may not always be necessary. Additionally, reevaluating the requirement for pressure injectors in curative treatments is important. These injectors are costly and demand skilled supervision to prevent damage to palm tissues, which can lead to palm mortality if the pressure exceeds 2 bar. Instead, adopting a standardized approach that combines mechanical sanitization with the simple diffusion technique of ‘drill and inject’ is recommended. The removal and appropriate disposal of severely infested palms present significant challenges. In numerous countries, the use of expensive shredding machines, which mandate trained personnel for operation, hinders the implementation of this aspect of the strategy. Moreover, there is a risk of weevil escape during the transportation of infested palms to a shredder located outside the farm or at a designated shredding site. Ferry [133] suggests exploring the possibility of processing or disposing severely infested palms directly on farms using compact portable shredders. This approach should be further investigated. Area-wide RPW IPM programs generate a substantial amount of data that necessitate collection, processing, and analysis. Regular validation and performance analysis of the control program are critical for obtaining insights into the situation and making efficient resource allocation decisions. Addressing this challenge is of utmost importance. Accurate record-keeping plays a vital role in the meaningful validation of the control program, particularly in terms of weevil capture in traps and the number and locations of infested palms, treated palms, and eradicated palms. Field maps of operational areas can facilitate record-keeping by enabling the plotting of trap positions and infested palms. Assigning unique numbers to each trap in the field and geo-referencing all traps and reported infestations are essential practices. If dedicated Geographic Information System (GIS) specialists are unavailable, individuals can create spatial and temporal spreadsheets periodically by plotting weevil captures in traps and marking infested palms on maps using distinct colors. High weevil captures or the removal of a significant number of infested palms indicate the need for adjustments in the strategy. Implementing a systematic data collection process linked to a GIS is crucial. By utilizing numerous mapped traps and recorded weevil captures, it becomes feasible to monitor temporal and geographical changes in pest distribution and effectively locate infested palms detected using various methods [130].

In the future, the development and validation of mobile apps for smartphones have the potential to greatly enhance data collection, compilation, and analysis in area-wide RPW IPM programs. FAO has recently commenced the validation process for the SusaHamra app, with the objective of aiding farmers in proficiently monitoring and managing RPWs. Furthermore, efforts are underway to establish a global platform that maps field data and provides analytics to facilitate improved decision-making. At level 2 of the strategy, the implementation of phytosanitation and quarantine measures plays a crucial role. However, there are several gaps and challenges that need to be addressed: Insufficient implementation of national/regional phytosanitary and quarantine regulations specifically targeted at the RPW; consistency in treatment protocols for palms before transportation and upon arrival at their intended destination; weak enforcement of regulations due to a lack of adequately trained personnel; difficulties in accessing certified planting material [129]. FAO (2020) recently addressed concerns regarding RPW management in the FAO guidelines [133]. While chemical protocols exist for quarantine purposes, there is a need to develop an effective treatment protocol specifically for the large palms used in ornamental gardening before transportation. The responsibility for implementing phytosanitary measures against the RPW lies with the respective national plant protection organizations. One of the significant challenges is the absence of field-worthy biological control agents for the RPW. Although some biological control agents are known, their delivery to the target site within the palm and their long-term sustainability require attention. Additionally, the importance of adopting optimal agrotechniques, including managing palm density, frond and offshoot removal, and irrigation practices, is often underestimated.

Further research is critical to establish a quantitative relationship between these factors and the incidence and severity of RPW infestations. Understanding host plant resistance against the RPW is currently limited, presenting a new area for exploration. Both traditional plant breeding techniques and advanced molecular-based methods can be employed to confer resistance against the RPW in common palm cultivars. The complete genome sequencing of the date palm cultivar ‘Khalas’ offers an opportunity to integrate genetic engineering into date palm breeding programs, enabling the overcoming of current limitations and the incorporation of desirable traits such as yield, quality, and resistance to abiotic and biotic stresses [134,135,136]. In many countries, RPW IPM programs are predominantly implemented by the state with minimal or no participation from farmers. This lack of farmer involvement hampers the effectiveness and success of the strategies. It is crucial to develop mechanisms that encourage farmers’ active participation in RPW IPM programs. Abdedaiem et al. [137] emphasized the importance of socio-economic studies to enhance farmers’ engagement in the RPW control program. Recent publications have focused on cutting-edge molecular aspects of the RPW, including RNAi, gene expression, and related studies [138,139]. The findings from these studies should be utilized to improve the control strategy. Faleiro and colleagues [129] emphasized the presence of several new tools for RPW IPM in the market. These tools include detectors, surveillance drones, pesticides, palm injectors, semiochemicals, biological control agents, palm shredders, and microwave treatment devices. It is essential to conduct proper testing and validation of these IPM tools at national and regional levels to ensure the accessibility of cost-effective and user-friendly technologies for farmers.

## 6. Conclusions

The RPW remains a significant global threat to palm tree survival. The primary route of RPW introduction into new areas is through the international trade and transportation of infested palm planting material for plantations and landscaping. Current RPW management strategies focus on monitoring and mass trapping with pheromones, implementing agronomic and phytosanitary measures, and utilizing biological control methods to some extent. IPM programs for the RPW also emphasize capacity building and quarantine measures. However, despite substantial global efforts, there are notable challenges and gaps in these management strategies that need attention. These challenges encompass early infestation detection, optimizing pheromone-baited traps, removing highly infested palms, reducing reliance on excessive insecticide use, and increasing farmer involvement in control efforts. Moving forward, the future prospects for RPW management may involve validating management programs, testing advanced technologies for practical field application, and exploring the potential of RNAi technology in control programs. It is important to recognize that effectively managing RPWs in the field is a complex undertaking. Nevertheless, with sufficient resources, appropriate interventions, and robust coordination, planning, and financial support, effective control of this pest can be achieved using existing technologies.

## Figures and Tables

**Figure 1 insects-14-00859-f001:**
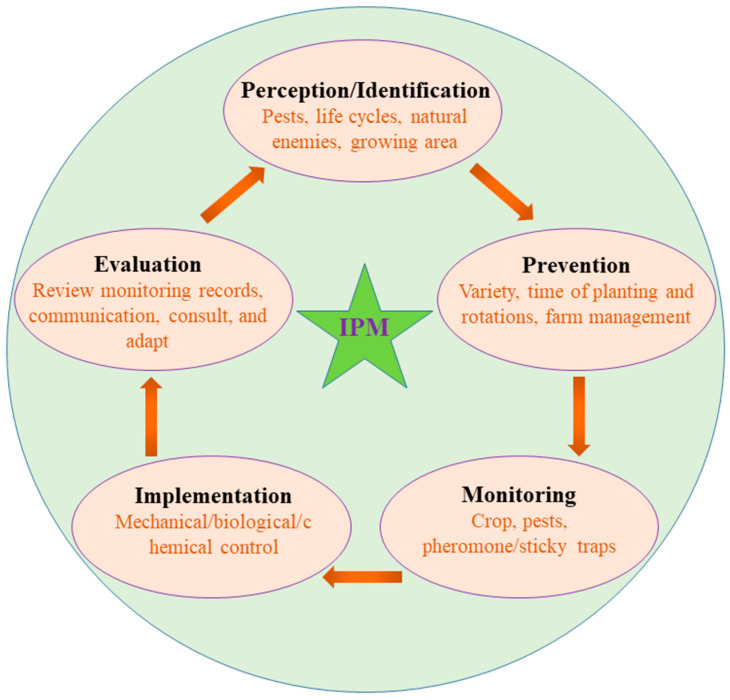
Integrated pest management strategies.

**Figure 2 insects-14-00859-f002:**
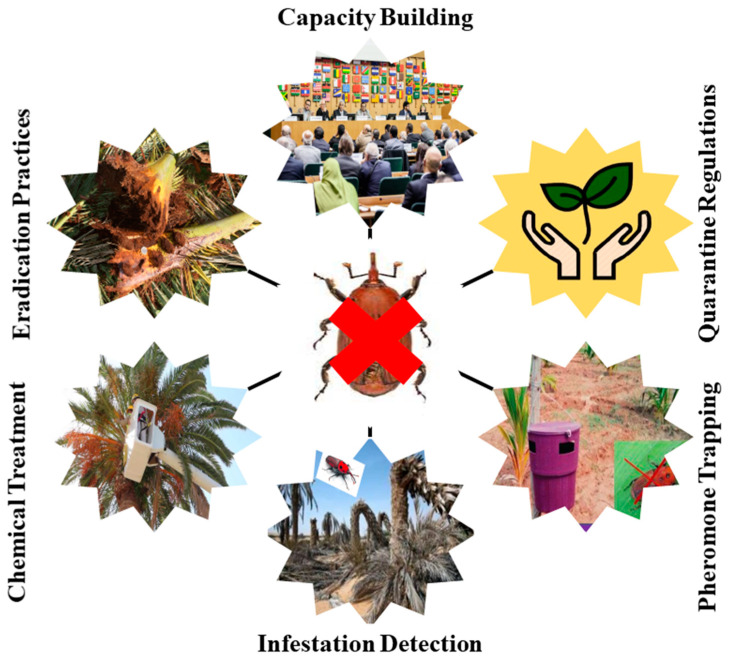
Red palm weevil management strategies.

## Data Availability

Not applicable.

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
