# Peer review of "Sustainable Pest Management in Date Palm Ecosystems: Unveiling the Ecological Dynamics of Red Palm Weevil (Coleoptera: Curculionidae) Infestations"

_insects, 2023, doi:10.3390/insects14110859_

Round 1
Reviewer 1 Report
Comments and Suggestions for Authors
Dear Authors
The validity of the topic raised in the article sent for review as much as predestinates it for publication. However, as a review article, it does not make an original contribution. Its function, however, is to interest the reader in the problem raised and to familiarize him as best as possible with the state of research on the harmfulness and methods of combating RPW. However, in this light the article has some shortcomings that need to be removed before publication, especially in a high-class journal like Insects.
The presented article is interesting, but does not answer a number of questions. Instead, it is full of general statements – right but trivial. Many paragraphs contain the same thought or statement repeated in different forms. This gives the impression that the article is popular scientific and does not correspond to the post of the journal to which it was sent. Giving more facts, details, concrete examples would be more desirable for the reader – a professional entomologist. So instead of general statements, it would be welcome to saturate the article with concrete.
It is also necessary to remove editorial errors: repetition of sentences, the correct form of applying shortcuts, explanations and calculations.
I am not convinced of the need to include figures 1 and 2. They do not contribute anything to the content and do not improve the perception of the article.
More details at the edge of the text.

Author Response
Dear Reviewer,
I trust this message finds you well. I want to express my sincere gratitude for your invaluable comments and insightful suggestions, which have significantly contributed to the enhancement of our manuscript. We have taken into consideration all of your suggestions and implemented the recommended changes, resulting in a refined version of the manuscript. As part of this process, we have removed Figure 1 and Figure 2 and replaced them with a more appropriate figure. Furthermore, we carefully considered all your comments and thoroughly included your suggestions in the main document. For your convenience, we have compiled our responses to your comments and suggestions in the attached document. Thank you once again for your time and assistance.
Best regards,

Reviewer 2 Report
Comments and Suggestions for Authors
The manuscript provides a comprehensive review of the red palm weevil problem and strategies needed to manage the pest. This review should be published. I included comments and suggestions in an annotated copy uploaded with this review. I noticed some repeated sentences within paragraphs, and topics tend to be repeated throughout the manuscript. I think it could benefit from a rigorous proof-reading/ editing by the authors.

Author Response
Dear Reviewer,
I trust this message finds you well. I want to express my sincere gratitude for your invaluable comments and insightful suggestions, which have significantly contributed to the enhancement of our manuscript. We have taken into consideration all of your suggestions and implemented the recommended changes, resulting in a refined version of the manuscript. Furthermore, we carefully considered all your comments and thoroughly included your suggestions in the main document. For your convenience, we have compiled our responses to your comments and suggestions in the attached document. Thank you once again for your time and assistance.
Best regards,

Reviewer 3 Report
Comments and Suggestions for Authors
It seems that this manuscript is very important as a revue for this insect species, red palm weevil, Rhynchophorus ferrugineus, that affects palm cultivar in Asia, Europe and certain areas of Central America. But in South America there is a similar especies affecting Oil Palm cultivars, Rhynchophorus palmarum, that causes a very serious problem to oil palm cultivars.
Annex file with some personal publications on R. palmarum, considering that authors should mention in this document.
I consider that this review on R. ferrugineus, should be published.

Author Response
Dear Reviewer,
I trust this message finds you well. I want to express my sincere gratitude for your invaluable comments and insightful suggestions, which have significantly contributed to the enhancement of our manuscript. We have taken into consideration all of your suggestions and implemented the recommended changes, resulting in a refined version of the manuscript. As part of this process, we have included the text and references to the main document. Thank you once again for your time and assistance.
Best regards,
Round 2
Reviewer 1 Report
Comments and Suggestions for Authors
Dear Authors
The current version of your manuscript is much better than the previous one. After removing one inaccuracy (see text), the manuscript can be forwarded for further processing.
Yours faithfully

Author Response
Dear Reviewer,
Many thanks for the suggestions. We went through the recommended article and improved the designated inaccuracy in the main file.
Best Regards,
Corresponding author